# Antifibrotic Effect of Selenium-Containing Nanoparticles on a Model of TAA-Induced Liver Fibrosis

**DOI:** 10.3390/cells12232723

**Published:** 2023-11-28

**Authors:** Elena G. Varlamova, Michail Victorovich Goltyaev, Vladimir Vladimirovich Rogachev, Sergey V. Gudkov, Elena V. Karaduleva, Egor A. Turovsky

**Affiliations:** 1Institute of Cell Biophysics, the Russian Academy of Sciences, Federal Research Center “Pushchino Scientific Center for Biological Research of the Russian Academy of Sciences”, 142290 Pushchino, Russia; goltayev@mail.ru (M.V.G.); vladimirrogachev6@gmail.com (V.V.R.); e.v.karaduleva@gmail.com (E.V.K.); turovsky.84@mail.ru (E.A.T.); 2Prokhorov General Physics Institute, the Russian Academy of Sciences, 119991 Moscow, Russia; s_makariy@rambler.ru; 3Department of Biophysics, Lobachevsky State University of Nizhny Novgorod, 603022 Nizhny Novgorod, Russia

**Keywords:** selenium, sorafenib, selenium nanoparticles, liver fibrosis, apoptosis

## Abstract

For the first time, based on the expression analysis of a wide range of pro- and anti-fibrotic, pro- and anti-inflammatory, and pro- and anti-apoptotic genes, key markers of endoplasmic reticulum stress (ER-stress), molecular mechanisms for the regulation of fibrosis, and accompanying negative processes caused by thioacetamide (TAA) injections and subsequent injections of selenium-containing nanoparticles and sorafenib have been proposed. We found that selenium nanoparticles of two types (doped with and without sorafenib) led to a significant decrease in almost all pro-fibrotic and pro-inflammatory genes. Sorafenib injections also reduced mRNA expression of pro-fibrotic and pro-inflammatory genes but less effectively than both types of nanoparticles. In addition, it was shown for the first time that TAA can be an inducer of ER-stress, most likely activating the IRE1α and PERK signaling pathways of the UPR, an inducer of apoptosis and pyroptosis. Sorafenib, despite a pronounced anti-apoptotic effect, still did not reduce the expression of caspase-3 and 12 or mitogen-activated kinase JNK1 to control values, which increases the risk of persistent apoptosis in liver cells. After injections of selenium-containing nanoparticles, the negative effects caused by TAA were leveled, causing an adaptive UPR signaling response through activation of the PERK signaling pathway. The advantages of selenium-containing nanoparticles over sorafenib, established in this work, once again emphasize the unique properties of this microelement and serve as an important factor for the further introduction of drugs based on it into clinical practice.

## 1. Introduction

Liver fibrosis is a dynamic process and the result of a sustained wound-healing response to chronic injury to the organ and involves interactions between hepatocytes, hepatic stellate cells, sinusoidal endothelial cells, and immune cells. Morphologically, liver fibrosis is characterized by the accumulation of extracellular matrix and the formation of a fibrous scar, which destroys the physiological architecture of the liver [1]. This is accompanied by the loss of hepatocytes and dysregulation of normal liver function, ultimately leading to liver failure [2]. Liver fibrosis is a reversible process—that is, when the agent causing the fibrotic reaction is removed, regression of fibrosis is possible [3,4]. However, if not treated promptly, it can lead to progressive liver cirrhosis and hepatocellular carcinoma (HCC). Fibrogenesis is activated by the proliferation of myofibroblasts, the main source of which is liver stellate cells, as well as endogenous portal fibroblasts, fibrocytes, bone marrow cells, and liver parenchymal cells [5].

One of the most studied drugs used to treat fibrosis of various organs is the multikinase inhibitor sorafenib, which is approved by the FDA for the treatment of hepatocellular and renal cell carcinoma [6,7,8]. However, the small-molecule drug sorafenib leads to serious side effects, the main reason for which is the nonspecific uptake of the drug into normal tissues. This results in hypertension, diarrhea, hand–foot syndrome, and other side effects [9,10]. In addition, sorafenib is characterized by poor solubility and absorption efficiency in the gastrointestinal tract. This is a good reason for the development of drugs for targeted delivery to the liver, which have high bioactivity and low toxicity for normal tissues and organs.

In recent decades, selenium nanoparticles and their various modifications have become increasingly popular as an effective drug carrier [11,12,13,14,15,16,17,18,19]. Very often, selenium nanoparticles (SeNPs) are used in combination therapy, which increases the effectiveness of treatment, and also as carriers of chemotherapeutic agents such as cisplatin [20,21], 5-fluorouracil [22], doxorubicin [23], and irinotecan [24], which demonstrates the presence of a synergistic effect between anticancer drugs and Se. When studying the effect of SeNPs in combination with radiation therapy using the example of non-small cell lung cancer, one of the most common cancers in the world, a decrease in cell proliferation, migration, invasion, and apoptosis was shown [25]. In particular, there are works in which SeNPs were used for the treatment of hepatocellular carcinoma. Therefore, when a sorafenib nanocomplex and selenium nanoparticles loaded in poly (d, l-lactic acid-co-glycolic acid)-b-poly (ethylene glycol) -b-poly (d, l-lactic acid-co-glycolic acid) were used, controlled, sustained drug release into HepG2 hepatocellular carcinoma cells was observed [26]. Using this cell line as an example, we also tested the effect of the sorafenib nanocomplex and selenium nanoparticles and carried out a comparative analysis of the cytotoxic effect of this nanocomplex separately with the effect of sorafenib and selenium nanoparticles on cells. The work established that doping SeNPs with the active compound sorafenib leads to an increase in its anticancer properties and induction of the early stages of apoptosis, which was not observed when HepG2 cells were treated separately with sorafenib or bare selenium nanoparticles [27]. However, there are practically no studies devoted to the study of selenium nanoparticles or selenium-based nanocomplexes for the treatment of liver fibrosis, which is extremely important, since it is chronic fibrosis that leads to hepatocellular carcinoma.

This work is the first to conduct a comprehensive study of the antifibrotic properties of SeNPs in comparison with sorafenib (So) and selenium and sorafenib nanocomplex (SeSo) in an animal model. We developed a protocol for the treatment of C57BL/6J mice with liver fibrosis caused by intraperitoneal injections of thioacetamide (TAA), various concentrations of selenium nanoparticles, and in combination with sorafenib.

## 2. Materials and Methods

### 2.1. Materials

Thioacetamide 98%, (Sigma-Aldrich, Burlington, MA, USA, #172502), qPCRmix-HS SYBR (Evrogen Moscow, Russia #PK147S), an MMLV RT kit (Evrogen, Moscow, Russia #SK021), ExtractRNA (Evrogen, Moscow, Russia #BC032), DNaseI (Invitrogen, Carlsbad, California, #18047019), Picro sirius red (Abcam, Cambridge, UK #ab246832), hematoxylin–eosin (HistoPoint, Saint Petersburg, Russia), and sorafenib (Bayer HealthCare AG, Leverkusen, Germany) were procured. The following reagents from Evrogen, Moscow, Russia, were used in this work: synthesis of gene-specific oligonucleotides, ExtractRNA reagent (#BC032), an MMLV-RT kit for cDNA synthesis (#SK021), a qPCRmix-HS SYBR mixture for PCR in real time (#PK147S), and DNA length markers (#NL001, #NL002). The following primary antibodies were used in the work: anti-GAPDH (Thermo FS, Waltham, MA, USA #14-9523-82), anti -CHOP (Thermo FS, Waltham, MA, USA #MA1-250), anti-CASP-3 (Abcam, Cambridge, UK#ab184787), anti-CASP-12 (Abcam, Cambridge, UK #ab235180), anti-IL-1β (Abcam, Cambridge, UK #ab205924), anti-IL-6 (Abcam, Cambridge, UK #ab290735), anti-IL-33 (Abcam, Cambridge, UK#ab187060), anti-TNFα (Abcam, Cambridge, UK #ab307164), anti-α-SMA (Abcam, Cambridge, UK #ab7817), anti ATF-4 (Abcam, Cambridge, UK #ab216839). Rabbit-anti-mouse Ab (Abcam, Cambridge, UK #ab6728) and mouse anti-rabbit Ab (Abcam, Cambridge, UK #ab 99697) were used as secondary antibodies. PVDF membranes (Thermo FS, Waltham, MA, USA #LC2005) were used for protein transfer. Protein concentrators (Thermo FS, Waltham, MA, USA #88517) were used to concentrate proteins in liver lysates. Histological sections were performed using a Thermo Scientific Microm HM 325 microtome (Thermo FS, Waltham, MA, USA). To fix stained histological sections, synthetic mounting medium Vitrogel (BioVitrum, Saint Petersburg, Russia) was used. To characterize the nanoparticles, a galvanomechanical scanner LScanH (Ateko-TM, Moscow, Russia), a Zetasizer Ultra Red Label (Malvern, Worcestershire, UK), and a transmission electron microscope 200FE (Carl Zeiss, Oberkochen, Germany) were used. The fluorescence of the samples was studied on a spectrometer FP-8300 (JASCO Applied Sciences, Halifax, Canada) and a precision multi-wavelength digital refractometer Abbemat MW (Anton Paar, Graz, Austria). Analysis of ALT and AST activities was carried out using commercial kits (# E-BC-K235-M and #E-BC-K236-M, respectively, Proteins-Antibodies, Moscow, Russia) and using an iMark™ Microplate Absorbance Reader (Biorad, Hercules, CA, USA).

### 2.2. Animals

This work used male mice of the C57BL/6J line (weight 15 g, 2–3 weeks old), which were purchased from the Stolbovaya branch of the Federal State Budgetary Institution of Science “Scientific Center for Biomedical Technologies of the Federal Medical and Biological Agency” of Russia. All animals were certified in accordance with the regulations on quality control of laboratory animals, nurseries, and experimental biological clinics (vivariums).

### 2.3. Injection Protocol

To induce liver fibrosis, male C57BL/6J mice (weight 15 g, 2–3 weeks) were intraperitoneally injected with TAA (150 µg/g mouse weight) twice a week for three months. After that, the animals were divided into several groups and administered either sorafenib (1 µg/g and 5 µg/g mouse weight), selenium nanoparticles (1 µg/g and 5 µg/g mouse weight), or a nanocomplex of sorafenib and selenium nanoparticles (1 µg/g and 5 µg/g mouse weight). Injections of nanoparticles were carried out intraperitoneally for a month; the indicated concentrations of nanoparticles were administered to the mice every other day. The results of the treatment for liver fibrosis were analyzed one and a half months after treatment. One group of animals was self-healing during the entire period of injection with nanoparticles. The control group consisted of animals that received injections of saline solution in the same volumes as in the experimental groups. The number of animals in each experimental group was 10.

### 2.4. Method for Obtaining Selenium Nanoparticles

Selenium nanoparticles (SeNPs) and sorafenib-doped selenium nanoparticles (SeSo) were prepared by laser ablation of bulk selenium targets with a polished top surface in deionized water. To do this, water was added to the cuvette at the bottom of which the target was located, covering the target by 2–3 mm, after which the massive target was irradiated with a laser beam (λ = 532 nm; T = 8 ns; f = 10 kHz; P = 20 W; Ep = 5 mJ). The displacement of the laser beam on the target along a given path in the form of parallel straight lines inscribed in a square with a step of 10 μm was carried out using an LScanH galvanomechanical scanner [28]. By changing the characteristics of laser radiation, the mixing speed, and the trajectory of the laser beam, it is possible to control the geometric parameters of nanoparticles: their size, concentration in the colloidal solution, and electro-kinetic potential, which was carried out using a Zetasizer Ultra Red Label [29]. Sorafenib was dissolved in 0.1 M citrate buffer (pH 4.1) to a final concentration of 30 mg/mL, since the solubility of sorafenib in water depends on the pH. Selenium nanoparticles were added to an aqueous solution of sorafenib in citrate buffer and incubated for 30 min. After this, to separate the nanoparticles from the initial solution, centrifugation was carried out using a Sigma 3-18KS centrifuge at a temperature of 4 °C and a rotor speed of 18,000 rpm with the Spincontrol S control system turned on [30]. This procedure was carried out three times. The amount of sorafenib associated with the nanoparticles was determined by the change in the absorption spectrum of the solution before the addition of the nanoparticles and after the deposition of the nanoparticles. A preparation of selenium nanoparticles with a concentration of 10^11^ × mL^−1^ can contain about 12 mg/mL of sorafenib on the surface of the nanoparticles.

The morphology of the nanoparticles was studied using a 200FE transmission electron microscope. The fluorescence of the samples was studied on an FP-8300 spectrometer. The measurements were carried out with the shutter turned on in quartz cells with an optical path length of 10 mm at room temperature (~22 °C). Each sample was measured three times. The refractive index of the media was measured using an Abbemat MW precision multi-wavelength digital refractometer. The experimental approaches used in the measurements were published previously [31].

### 2.5. RNA Isolation, Reverse Transcription, RT-PCR

RNA isolation was carried out using the ExtractRNA reagent (Evrogen), intended for the isolation of total RNA from biological samples. This reagent is a monophasic solution of phenol and guanidine isothiocyanate. Liver tissue was homogenized by pipetting in 1 mL of ExtractRNA reagent, and then total RNA was isolated according to the manufacturer’s protocol. The quality of RNA isolation was checked using electrophoresis in a 1% agarose gel as well as using a spectrophotometer at a wavelength of 260 nm. To prevent contamination of the RNA samples with genomic DNA, they were treated with DNase I at 37 °C for 1 h, after which the enzyme was inactivated by adding 50 mM EDTA to the mixture and heating to 60 °C for 10 min.

The reverse transcription reaction was carried out according to the protocol and using a first-strand cDNA synthesis reagent kit containing murine leukemia virus (MMLV) reverse transcriptase. The content of total RNA (0.5–2 μg) was controlled by performing a parallel amplification reaction using primers specific to the reference gene.

The resulting cDNA was used as a template for RT-PCR using the qPCRmix–HS SYBR mixture containing the intercalating dye SYBR Green I. The amplification reaction was carried out at the following temperature conditions: 95 °C for 1 min; 95 °C for 10 s, 60 °C for 10 s, and 72 °C for 15 s (35 cycles). The relative level of gene expression (RUE—the level of expression of the gene under study relative to the expression of the reference gene) in each cell line was determined by the formula. The change in the level of mRNA expression of the studied proteins before and after treatment was determined by the formula TUE = 2^−ΔΔ Ct^, where ΔΔCt is the difference in ΔCt values for each gene before and after cell treatment. Each experimental cycle was repeated three or more times. When performing RT-PCR, reference gene-encoding glyceraldehyde-3-phosphate dehydrogenase was used. The sequences of all primers used in RT-PCR are given in Table 1.

### 2.6. Western Blotting

Liver samples were homogenized in lysis buffer (20 mM Tris, 150 mM NaCl, 2 mM EDTA, 1 mM PMSF, 1% Triton-X100), incubated on ice for 15 min, and then centrifuged for 30 min at 20,000 g at 4 °C, and the supernatant fraction was concentrated. A total of 70 μg of protein was added per well for each sample. The proteins were separated by PAGE electrophoresis in a 12.5% polyacrylamide gel, after which the proteins were electro-transferred to a PVDF membrane. The membranes were blocked in 5% BSA for 5 h at room temperature or for 15 h at 4 °C, and then the membranes were incubated with primary antibodies in 3% BSA for 2 h at room temperature or for 15 h at 4 °C. After thoroughly washing the membranes in 1x PBST, they were incubated with secondary antibodies for 2 h at room temperature and then washed thoroughly, and the immunoreactive bands were visualized by determining the peroxidase activity using DAB staining (0.05% DAB in 1× PBS + 10 μL 30% peroxide hydrogen).

### 2.7. Histological Analysis

Liver samples were fixed with a solution of 10% formalin and 0.05% glutaraldehyde in 0.9% NaCl and passed through solutions of ethyl alcohol in increasing order of concentration and *o*-xylene. Then, the samples were embedded in paraffin and sections were cut at a thickness of 3 μm using a Thermo Scientific Microm HM 325 microtome (Thermo FS, Waltham, MA, USA). The sections were deparaffinized using *o*-xylene and ethyl alcohol solutions in descending order of concentration, and then the sections were stained with solutions of dyes—hematoxylin-eosin and picro-sirius red—according to the staining protocols of the specified kits. The stained sections were fixed on slides using the synthetic mounting medium Vitrogel. The sections were photographed using a Leica DM6000B microscope. To calculate the relative area of collagen fibers, the ImageJ program was used according to the specified calculation method: https://imagej.nih.gov/ij/docs/examples/stained-sections/index.html (accessed on 15 January 2023).

### 2.8. Measurement of ALT/AST Activities

Analysis of enzyme activities in serum was performed using the Reitman–Frankel colorimetric method. To do this, the animals were first sedated with a solution of Rometar/Telazole (5 mg/kg and 50 mg/kg, respectively), after which the animals were killed by decapitation. Blood samples were incubated for 30 min at room temperature until the blood clotted, and the resulting clot was centrifuged at 2000 g for 15 min at 4 °C. Serum was collected and used to analyze the activities of ALT and AST enzymes using commercial reagent kits. Briefly, a specific amount of aspartate (AST/GOT) or alanine (ALT/GPT) was added to the sample and incubated for 30 min at 37 °C, and then phenylhydrazine was added to stop the enzymatic reactions and the samples were incubated for 20 min at 37 °C, after which the 0.4 N NaOH solution was added and incubated with it for 10 min at room temperature. Next, optical measurements of the visible color absorption of phenylhydrazone were carried out on a microplate reader at a wavelength of 490 nm.

### 2.9. Statistical Data Processing

Microsoft Excel and GraphPadPrism 5 software were used to analyze the data, generate graphs, and process the statistics. Protein assessment was carried out in different samples using the Lowry method. Protein concentration was calculated using a standard curve constructed using a 1 mg/mL BSA solution. Values are given here as the mean ± standard deviation of at least three independent experiments. Differences were considered significant at *p* < 0.05. Protein expression was quantified using ImageJ software, version number 1.8.0. Origin 8.5 (Microcal Software Inc., Northampton, MA, USA) and Prism 5 (GraphPad Software, La Jolla, CA, USA) were used for plotting and statistical processing, respectively. The significance of differences between experimental groups was determined using analysis of variance (one-way ANOVA with post-hoc Tukey test or post hoc Student–Newman–Keuls test) or Student’s *t*-test, and within groups, Student’s *t*-test. Differences were considered significant as follows: *** for *p* < 0.001, ** for *p* < 0.01, * for *p* < 0.05; n/s—differences were not significant. To calculate the relative area of collagen fibers, the ImageJ program was used according to the specified calculation method: https://imagej.nih.gov/ij/docs/examples/stained-sections/index.html (accessed on 15 January 2023). 

## 3. Results and Discussion

The results presented in this study indicate for the first time a direct connection between liver fibrosis induced by TAA injections and the development of endoplasmic reticulum stress (ER-stress), which is accompanied by the activation of both adaptive unfolded protein response (UPR) signaling pathways and apoptotic ones. In addition, these processes are associated with inflammation. The selenium-based nanoparticles developed in this work have a number of significant advantages aimed at preventing these negative consequences.

### 3.1. General Characteristics of the Obtained Selenium Nanoparticles

A preparation of selenium nanoparticles with a monomodal size distribution and particle concentration of 10^11^ × mL^−1^ was obtained. The average hydrodynamic diameter of SeNPs was about 100 nm, with a half-width in the range of 70–130 nm (Figure 1A). After the attachment of So to SeNPs, an increase in the hydrodynamic diameter of the SeSo complex by 10–20 nm was observed. The average zeta potential of SeNPs was in the order of −30 mV, whereas the zeta potential of the SeSo complex was in the order of −20–25 mV (Figure 1B). It should be noted that the aqueous colloidal solution of SeNPs had a weak reddish tint and absorbed rather weakly in the wavelength range of 240–300 nm, whereas the SeSo nanocomplex absorbed intensely in the wavelength range of 240–300 nm. The absorption spectrum of SeSo was quite characteristic and had four local maxima. This spectrum was imparted to the complex by sorafenib molecules, but the nature of the spectrum did not change qualitatively when sorafenib was added to the selenium nanoparticles (Figure 1C). According to electron microscopy data, the nanoparticles had a spherical shape (Figure 1D). TEM is equipped with an attachment for energy-dispersive X-ray spectroscopy. Using this method, it was shown that the nanoparticles consisted of selenium in the zero-valent state.

It was shown that at all wavelengths used for measurements, the refractive index was minimal for SeNPs and at the maximum for the SeSo complex. Moreover, the refractive index of the SeSo nanocomplex was higher than that of a solution of chemically pure So and a colloidal solution of SeSo in the same concentration (Figure 2).

When studying the fluorescence of the samples, it was shown that the colloidal solution of SeNPs almost did not fluoresce (Figure 3A), whereas the aqueous solution of So fluoresced intensely when excited in the wavelength range of 230–280 nm and emission was observed in the range of 275–300 nm. Also, there were two separate weakly defined emission peaks at 405 nm. Fluorescence maxima were observed upon excitation at 266 nm and 236 nm (Figure 3B). The fluorescence of the SeSo complex appeared to be due to the fluorescence of sorafenib (Figure 3C). The fluorescent spectrum was similar but had almost two times less intensity.

### 3.2. The Weight of Animals Returns to Normal when Injected with Selenium-Containing Nanoparticles

When comparing the body weight of mice in control groups with the weight of mice after injections of TAA (150 µg/g) for three months, it was seen that there was a noticeable decrease in the body weight of animals in the experimental group by an average of 2.7 g compared to the control group (Figure 4A), whereas the average weight of the liver in animals of both groups was the same (Figure 4B). This indicates that the ratio of liver weight to animal weight increased by an average of 12% after TAA injections relative to the control (Figure 4C).

Measurements of the average weight of animals and their livers after a course of injections with the studied nanoparticles did not reveal significant differences from the average weight of animals in the control group, with the exception of mice receiving injections of So at a concentration of 5 µg/g, for which the average weight of the animals increased by 4 g less compared to control. In addition, the average weight of the mice undergoing self-healing was 3 g lower than that of the control group (Figure 4D).

When comparing the liver weight of animals in the self-healing group and in the groups after nanoparticle injections with the control group, we can conclude that mice from the self-healing groups, as well as those receiving injections of 5 µg/g So and SeSo, differed the most from the control (Figure 4E). The liver weight of these animals was 0.3 g lower than the average liver weight of the control group. However, the ratio of liver weight to animal weight was, on average, 12% lower in the group of mice after injections of 5 µg/g SeSo, whereas in the other two groups this figure was close to the control. It is worth noting that the ratio of liver weight to animal weight in the group of mice injected with 1 µg/g SeNPs was, on average, 21% higher than in the control (Figure 4F).

Thus, the average weight of animals in the self-healing group and after injections of So at a concentration of 5 µg/g did not return to control values; however, the ratio of liver weight to total weight of the animals in these groups was, on average, close to normal. What was not observed in the groups of mice after injections of 1 µg/g SeNPs and 5 µg/g SeSo, was that the average the values of this indicator were higher by 21% and lower by 12%, respectively.

### 3.3. Selenium-Containing Nanoparticles Have an Antifibrotic Effect

Macroscopic analysis of liver samples, shown in Figure 5A,B, indicated numerous inclusions of connective tissue in the livers of mice treated with TAA, which indicates that there were pathological disturbances in the architectonics of the liver and the formation of fibrotic changes in it.

Subsequent microscopic analysis of histological liver samples confirmed the collagenous nature of the identified structures, which is consistent with literature data on the induction of fibrosis through the toxic effect of TAA [32,33,34]. Figure 6(Ab,Ac) shows that global architectural distortions occurred in the liver compared to the normal tissue architecture in the control, in which the liver plates radiated away from the central vein, the liver parenchyma was homogeneous and fine-grained, and there were no signs of fibrosis (Figure 6(Aa)).

In samples after TAA injections, fibrotic changes appeared as dark stripes along the tissue, dividing the liver into septa, which indicated a violation of the lobular structure of the liver, with the formation of false lobules of different sizes (1) surrounded by fibrous tissue (2). Dystrophy of the hepatocytes of the false lobules was pronounced. The described morphological changes were absent in the liver samples from all treatment groups, shown in photograph Figure 6(Ad).

To identify collagen structures in the liver, picro-sirius red was used, which selectively stained collagen types I and III and confirmed the collagen nature of the identified structures, as evidenced by the images in Figure 6B. Microscopy of preparations from the control group (normal liver), shown in Figure 6(Ba), showed red coloration only of the vessel walls, which is a normal site of collagen deposition.

After exposure to TAA, liver sections from the mice showed a wide range of morphological stages of the pathological process. Figure 6(Bb) shows mild focal portal and periportal fibrosis and extended portal triads. A pronounced concentric proliferation of collagen around the bile ducts (periductal fibrosis) was observed, which may indicate the formation of primary sclerosing cholangitis. Areas of moderately expressed venular and perivenular fibrosis were more common. At the same time, the structure of the liver parenchyma was disrupted; some fibrous septa were clearly shown to connect the centrilobular veins with the portal tracts, which are clearly visible in Figure 6(Bc). A severe degree of fibrosis was characterized by wide connective tissue septa forming the porto-portal and centro-portal septa, forming false lobules of the liver and regenerative nodules that disrupted the architectonics of the liver (Figure 6(Bd)). Thus, we observed a mix, the most common form of fibrosis, in which various stages of the fibrotic process were simultaneously represented.

In the treatment groups, a decrease in edema was observed (which was reflected in a more uniform lumen of the sinusoids), as well as restoration of the integrity of the hepatic beams. A significant number of regenerating hepatocytes in a state of mitosis was noted in the periportal zone, and the number of functionally active liver cells increased (Figure 7A,B). Compared with the TAA group, the pronounced inflammatory cell infiltration of the portal tracts was almost completely replaced by moderate diffuse infiltration of the liver parenchyma.

Subsequent experimental use of nanoparticles and sorafenib suggested a correction of the fibrotic state obtained by TAA. The area of the stained collagen was calculated using ImageJ, showing a significant increase in the TAA group, but the differences in the treatment groups were not significant (Figure 7C).

Thus, the consequences of induced severe toxic liver fibrosis in mice were significantly reduced with the use of nanoparticles and sorafenib, which indicates the antifibrotic effect of selenium nanoparticles, comparable to the biological effect of sorafenib.

### 3.4. Selenium-Containing Nanoparticles Reduce Cytolysis, Normalizing ALT and AST Levels in the Blood of Animals

Alanine aminotransferase (ALT) and aspartate aminotransferase (AST) are enzymes predominantly found in liver cells and the most important indicators of damage in this organ, so we analyzed the activity of these enzymes in the serum of the experimental animals.

According to the results shown in Figure 8A,B, the measurements of ALT and AST activity showed that TAA injections resulted in an increase in ALT and AST activity by more than 13 times relative to the control. In the blood of animals receiving injections of 1 and 5 µg/g SeNPs, a decrease in the activities of ALT and AST was observed of eight or more times compared to TAA injections. After injections of SeSo nanoparticles at concentrations of 1 and 5 µg/g, a decrease in ALT activity was observed of more than 9 and 11 times, respectively. Sorafenib at the same concentrations reduced ALT activity by 6.8 and 7.5 times, respectively, whereas AST enzyme activity was decreased by only 2.8 and 4.4 times compared to the group of animals injected with TAA.

Thus, selenium-containing nanoparticles significantly contributed to the normalization of the activity of ALT and AST enzymes in the blood of animals compared to sorafenib, which was an indicator of a decrease in liver cell damage (cytolysis) and the effectiveness of treatment with these selenium-containing agents.

### 3.5. Selenium-Containing Nanoparticles Have an Anti-Inflammatory Effect

At the first stage, it was important to establish the presence of fibrosis in the liver tissue of the studied animals. To do this, we analyzed a number of genes, the expression of which, according to previously obtained data, increases during fibrogenesis [32,33,34,35,36,37,38,39,40]. According to the results presented in Figure 9A, we can say that the TAA injection protocol we chose led to a fivefold increase in the expression of the mRNA of the TGF-β gene, which is a powerful stimulator of scar tissue accumulation, proliferation and activation of mesenchymal cells, and the biosynthesis of type 1 collagen [35,36,37,38]. In addition, the mRNA expression of the Col1a1 and Col1a2 genes, encoding pro-alpha 1 or 2 chains of type I collagen, increased by almost four and seven times, respectively. At the same time, an eightfold increase in the expression of alpha-smooth muscle actin (α-SMA) mRNA was observed, which indicates increased collagen synthesis [41].

Regarding the activation of hepatic stellate cells (HSCs), which is known to play a key role in the process of liver fibrogenesis and is accompanied by increased production of extracellular matrix (ECM) components by these cells [42], the PCR results we obtained most likely indicate the absence of their activation. Thus, the expression of mRNA of epidermal growth factor (EGF), which plays an important role in the regulation of HSC activation; glial fibrillar acidic protein (GFAP), a marker of early HSC activation [43]; and angiopoietin-1 (Angpt1), which is also responsible for HSC activation [44], decreased, and mRNA expression (platelet-derived growth factor (PDGF)) did not change (Figure 9A). This may be due to an almost 10-fold decrease in the mRNA expression of the NOX4 isoform of NADPH oxidases, the deficiency of which, along with NOX1 deficiency, protected the mice from the development of inflammation and liver fibrosis by inhibiting HSC activation [45,46].

In addition, it is known that various cytokines released by immune cells play a huge role in the regulation of liver fibrogenesis. IL-17 is secreted in fibrotic livers by T helper cells (Th17), neutrophils, and mast cells and is a pro-fibrotic cytokine that stimulates HSCs to increase levels of type I collagen, α-SMA, and TGF-β [47,48,49]. According to our results, we can state that there was an increase in IL-17 mRNA expression of two times compared to the control (Figure 9A). A number of studies have shown that high levels of IL-22 mRNA expression are observed in the blood of patients with cirrhosis and hepatocellular carcinoma of the liver [50,51], whereas IL-22 is able to inhibit liver fibrosis in mice [52]. Figure 9A demonstrates that TAA injections caused a strong inhibition of IL-22 mRNA expression (almost 5-fold), which may indirectly indicate the activation of fibrogenesis. In addition, TAA caused an almost fourfold increase in IL-33 mRNA expression, which is also evidence of fibrosis since this interleukin in mice is known to be released from damaged hepatocytes during fibrogenesis [53]. The level of IL-1β mRNA expression also tended to increase (twice compared to the control), which also indicates the presence of inflammation, since this interleukin, as a rule, has a pro-inflammatory effect on tissues and cells, which was also evidenced by the increased expression of IL-6 mRNA, one of the most important mediators of the acute phase of inflammation. It is known that IL-10 is a pleiotropic cytokine that has both anti-inflammatory and anti-fibrotic activity and has the opposite effect [54,55]. In our experiments, TAA caused a 2.5-fold increase in the expression of this interleukin mRNA. An active inflammatory response in the liver in response to TAA injections was confirmed by a strong increase in the expression level of CASP-1 mRNA, which plays a central role in cellular immunity as an initiator of the inflammatory response. The activation of inflammasome complexes is known to cleave pro-CASP-1 and trigger the maturation and secretion of IL-1β and IL-18, which are critical cytokines involved in immune responses and triggering the inflammatory cascade [56]. Recent studies have shown that inflammasomes and inflammation-related pyroptosis are involved in liver fibrogenesis in various pathologies [57]. Pyroptosis, which is a type of programmed cell death, depends on inflammatory caspase-1 to cleave gasdermins to form membrane pores [58]. Thus, increased CASP-1 mRNA expression can lead to pyroptosis of hepatocytes and immune cells [59]. According to our data, TAA promoted increased expression of pro-inflammatory cytokines TNF-α and INF-γ, secreted by macrophages and T cells, respectively. IL-17 is known to interact with TNF-α to increase the synergistic secretion of IL-6 in various cell types, including hepatocytes [60], which is also consistent with our data.

Thus, based on the expression patterns of a number of the above genes, we can conclude that the thioacetamide injection protocol we chose contributed to the development of severe liver fibrosis.

Further injections of SeNPs, So, and SeSo at concentrations of 1 and 5 µg/g led to a significant change in the expression patterns of most of the above genes (Figure 9). Thus, according to the results of real-time PCR, the level of expression of TGF-β mRNA decreased by 10 or more times, α-SMA by more than 5 times, and genes encoding chains 1 and 2 of type 1 collagen by an average of 5 times. A similar trend towards a decrease in expression levels was characteristic of the EGF and Angpt1 genes responsible for the activation of hepatic stellate cells. After injections of SeNPs, the expression of PDGF and GFAP mRNA remained virtually unchanged, whereas after injections of So and SeSo, an increase in the expression of the mRNA of these genes was observed.

When analyzing the expression patterns of interleukins, it was noted that the injections of nanoparticles caused a significant decrease in the expression of the mRNA of pro-inflammatory cytokines IL-1β, IL-6, IL-17, IL-22, IL-33, TNF-α, INF-γ, and CASP-1 (Figure 9E,I). Similar injections of So into mice did not affect IL-1β, IL-6, or IL-33 mRNA expression (Figure 9J). In addition, sorafenib injections caused an increase in the expression of PDGF and GFAP mRNA, returning them to normal levels. It is worth noting that NOX4 mRNA expression increased compared to the inhibitory effect of TAA on its mRNA expression after injections with all studied agents but without exceeding the control values (in the liver without TAA exposure).

During the progress of fibrosis, regions of hypoxia were observed in the liver, accompanied by increased expression of hypoxia-inducible factor 1α (HIF-1α) [61,62]. Although TAA did not significantly affect the expression of mRNA-encoding HIF-1α, subsequent injections of SeNPs, So, and SeSo slightly decreased this indicator.

We also checked the expression of STAT3 (Janus kinase (JAK)-signal transducer and activator of transcription (STAT)) mRNA, since STAT3 has been repeatedly shown to be involved in the regulation of fibrogenesis of various organs [63,64]. However, its role in these processes is still unclear, since it is involved in the regulation of various signaling pathways and can have both pro- and anti-inflammatory effects. In our studies, TAA did not significantly affect its expression.

The data we obtained regarding the expression of the mRNA of the genes under study were confirmed by the immunoblotting results presented in Figure 10.

Histological analysis revealed an anti-fibrotic effect in the self-healing group, slightly different from that in the groups with injections of SeNPs, So, and SeSo, which was confirmed by analyzing the expression of mRNA and the quantitative content of proteins α-SMA, Col1a1, Col1a2, EGF, IL-17, and IL-10 (Figure 9В,C). However, there are serious differences in the mRNA expression patterns of a number of anti- and pro-fibrotic and inflammatory genes and the quantification of the proteins they encode. Thus, the expression of TNF-α, PDGF, Angpt1, GFAP, IL-1β, IL-33, IL-22, IL-6, TNFα, and INF-γ mRNA in the livers of the animals from the self-healing group either did not change or increased more than twice as much as in the TAA group. This may indicate that, despite visually poorly noticeable differences, in the self-healing group, pro-fibrotic and pro-inflammatory processes were still preserved and even enhanced, which is supported by data on the relatively reduced weight indicators of the animals from this group compared to the control. The noticeable positive dynamics in the self-healing group can also be explained by the fact that liver regeneration, especially in initially healthy mice by nature, is very intense and that by removing the fibrosis inducer, the activation of the animal’s immune system improved a number of indicators. In particular, there was a decrease in collagen in the liver. However, this is extremely insufficient for complete recovery, especially against the backdrop of further growth of a number of pro-fibrotic and pro-inflammatory genes.

The results of real-time PCR and Western blotting regarding the expression of the above genes in the liver of mice before and after treatment are schematically presented in Figure 11.

Based on this scheme, it can be concluded that the number of genes whose mRNA expression levels returned to normal levels was greater after SeNP injections (14 genes), whereas after injections of So and SeSo there were only 10 genes. In the self-healing samples, the number of genes whose expression levels returned to control values was only 6, whereas the mRNA expression levels of 11 genes remained high.

In addition, based on the results of real-time PCR and immunoblotting that we obtained, we can assume the implementation of the following events in the liver when it is exposed to TAA (Figure 12). Exposure of liver cells to TAA leads to an acute inflammatory response accompanied by activation of caspase-1, formation of the inflammasome, and activation of IL-1β. These processes occur both in various immune cells and in hepatocytes, which, along with the growth of other pro-inflammatory cytokines, can lead to pyroptosis or cell necrosis. IL-1β, released into the extracellular environment, can contact receptors on the surface of liver stellate cells, activating them. This, in turn, is accompanied by excessive deposition of extracellular matrix and the growth of α-SMA, Col1a1, Col1a2, and EGF, which leads to liver fibrosis. Despite the well-observed trend towards a decrease in pro-fibrotic genes, after injections with sorafenib, high levels of expression of Casp-1 and IL-1β still remained, which increases the risks of maintaining cell pyroptosis, which was not observed after similar injections with nanoparticles containing selenium.

### 3.6. Selenium-Containing Nanoparticles Neutralize the Effects of Prolonged ER-Stress and Apoptosis in Liver Cells Caused by the Action of TAA

To understand the therapeutic effect of selenium-containing nanoparticles at the molecular level, we examined changes in mRNA expression patterns and the amount of protein they encode, primarily focusing on markers of ER-stress. We have repeatedly shown that various versions of selenium nanoparticles, as well as other organic and inorganic selenium-containing compounds, are capable of exerting cytotoxic effect in cancer cells through the regulation of the expression of these markers [14,27]. Such regulation is varied and is largely determined by the nature of the selenium-containing agents, the concentration used, and the origin of the cancer cells [65,66,67,68]. Based largely on our previous results, in this work, we checked how the mRNA expression of key markers of the PERK signaling pathway UPR (ATF-4), IRE1α signaling pathway UPR (XBP1s), and ATF-6 signaling pathway UPR (ATF-6) changed. In addition, the expression patterns of a number of genes that are involved in the adaptive response and pro-apoptotic signaling cascades associated with ER-stress were screened.

According to the real-time PCR results shown in Figure 13, we can conclude that TAA caused an increase in the expression of CASP-12, CASP-3, PUMA, CHOP, NRF-2, BAX and BAK, and JNK1 mRNA against the background of reduced mRNA expression of the anti-apoptotic genes BCL-XL and BCL-2. Considering that JNK1 expression increased almost fivefold, it is likely that IRE1α phosphorylated TRAF2 (TNF receptor-associated factor), triggering the TRAF2-ASK1-JNK1 signaling cascade [69]. An increase in the expression of the nuclear factor NRF-2 of six times may indicate activation of the PERK signaling pathway, since NRF-2 is known to be the target of this serine–threonine kinase [70].

It is known that activation of the caspase pro-apoptotic pathway during ER-stress occurs through two mechanisms. Firstly, it was found that, in mice under ER-stress, TRAF2 interacted with ER-stress-sensitive procaspase-12 [69], and secondly, that CASP-12 was activated under conditions of ER-stress, which was apparently carried out through the Ca^2+^ signaling pathway. When ER-stress occurs, conformational changes and/or oligomerization of pro-apoptotic proteins Bax and Bak (proteins belonging to the Bcl-2 family) occur on the ER membrane, which leads to damage to calcium stores in the ER and the release of Ca^2+^ into the cytosol. An increased Ca^2+^ flux activates m-calpain, which is a member of the family of Ca^2+^-dependent cysteine proteases [71]. Calpain, in turn, cleaves procaspase-12 to caspase-12, which leads to the activation of apoptosis. However, as part of this work, we did not check the activation of this pathway, so we can only say that, most likely, activation of caspase-12 occurred through TRAF-2 regulation.

To date, sufficient information has been accumulated regarding the complex regulation of CHOP activation, which affects various signaling pathways [72,73,74]. Summarizing our data regarding the expression of CHOP mRNA and the relative amount of the protein itself, it is most likely that the increase in its gene expression and relative amount in the liver cells after TAA injections was the result of the phosphorylation of p38MAPK or JNK1, as was previously demonstrated [75].

Subsequent injections of selenium-containing nanoparticles and sorafenib led to a sharp decrease in the levels of mRNA expression of the spliced form of the transcription factor XBP1s and ATF-6, key markers of the IRE1α and ATF-6 UPR signaling pathways, and a significant increase in the expression of mRNA of the transcription factor ATF-4, a key marker of the PERK signaling pathway UPR (Figure 13).

In addition, all agents decreased the mRNA expression levels of PUMA, CHOP, NRF-2, BAX, BAK, and, to varying degrees, JNK1. It is worth noting that only in the case of injections of SeNPs at both concentrations and SeSo at a concentration of 5 µg/g, respectively, was there a tendency towards a decrease in the expression of CASP-12 and CASP-3, which correlates with a decrease in the expression of JNK1 mRNA (Figure 13). In the case of injections with sorafenib, the expression levels and relative content of caspases CASP-12 and CASP-3 in liver cells did not change, and at a concentration of 5 µg/g even increased threefold against the background of high expression of JNK1, which indicates the progression of apoptosis in liver cells.

The data we obtained regarding the expression of the mRNA of the genes under study were confirmed by the immunoblotting results presented in Figure 14.

If we analyze the effects of So, SeNPs, and SeSo in relation to the normalization of mRNA expression and the amount of protein encoded by them in the liver, then we learn that SeNPs and SeSo nanoparticles normalized these indicators for 7 genes and So for 4 genes out of 17 studied in this part of the work (Figure 15).

Thus, based on our data, we can assume that apoptotic liver cell death is a consequence of exposure to TAA, which occurs even under conditions of prolonged ER-stress. We hypothesize the following scenario, presented in Figure 16: TAA causes activation of two of the three UPR pathways described so far, namely, IRE1α and PERK, since we observed increased expression of JNK1 and NRF-2, which are targets of the IRE1α and PERK kinases. Increased expression of PUMA and p53 leads to increased expression and translocation to mitochondria of the pro-apoptotic proteins BAK and BAX, which ultimately leads to apoptosome formation and activation of caspase-9 and caspase-3. In turn, active caspase-12 also activates effector caspase-3 through activation of caspase-9, which forms the apoptosome. Thus, under conditions of prolonged ER-stress caused by long-term action of TAA, pro-apoptotic signaling pathways are activated.

After So injections, liver cells still retained high levels of some pro-apoptotic genes: CASP-12, CASP-3, and JNK1, which may indicate a high risk of apoptosis in liver cells. On the other hand, after injections with nanoparticles and against the background of reduced expression of all pro-apoptotic genes, we can talk about the implementation of the adaptive UPR response through activation of the PERK signaling pathway, as evidenced by an increase in the expression level of its key target ATF-4. Thus, selenium-containing nanoparticles were dose-dependently able to neutralize the effects of ER-stress and lead to a high increase in the expression of mRNA of a number of pro-apoptotic genes caused by TAA injections.

## 4. Conclusions

We conducted a pilot study of the anti-fibrotic, anti-inflammatory, and anti-apoptotic effect of selenium-containing nanoparticles and performed a comparative analysis of these effects with the effect of the well-known drug sorafenib on a model of non-alcoholic mouse liver fibrosis caused by thioacetamide injections. Based on the analysis of the molecular regulation of fibrogenesis, we found that the two types of selenium nanoparticles we obtained (doped with and without sorafenib) led to a significant decrease in almost all pro-fibrotic and pro-inflammatory genes. Moreover, a comparative analysis of the expression patterns of these genes after injections of selenium nanoparticles and in control samples revealed a slight advantage of SeNPs.

Sorafenib injections also reduced mRNA expression of profibrotic and proinflammatory genes but less effectively than both types of nanoparticles. In addition, it was shown for the first time that TAA can be an inducer of ER-stress, most likely activating the IRE1α and PERK signaling pathways of the UPR. In this case, activation of the pro-apoptotic response to TAA was observed, which was accompanied by an increase in pro-apoptotic genes. Sorafenib, despite a pronounced anti-apoptotic effect, still did not reduce the expression of caspase-3 and -12, nor mitogen-activated kinase JNK1, to control values, which increases the risk of persistent apoptosis in liver cells.

After injections of selenium-containing nanoparticles, the negative effects caused by TAA were reversed, causing an adaptive UPR signaling response through activation of the PERK signaling pathway.

In general, we can conclude that the results obtained in this work allow us to speak with confidence about the high therapeutic effectiveness of selenium nanoparticles aimed at leveling liver fibrosis and the negative processes accompanying this disease. The advantages of selenium nanoparticles over sorafenib, established in the work, once again emphasize the unique properties of this microelement and serve as an important factor for the further introduction of drugs based on it into clinical practice.

## Figures and Tables

**Figure 1 cells-12-02723-f001:**
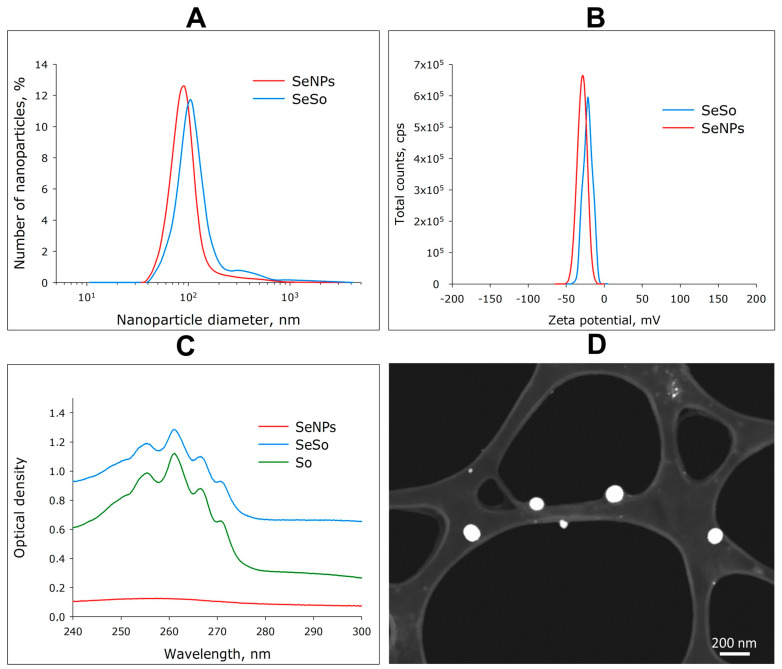
Basic physicochemical characteristics of a colloidal solution of selenium nanoparticles (SeNPs) and a complex of selenium nanoparticles with sorafenib (SeSo). (**A**) Hydrodynamic diameter of SeNPs and SeSo; (**B**) electro-kinetic potential of SeNPs and SeSo; (**C**) spectral properties of SeNPs and SeSo and chemically pure sorafenib (So); (**D**) TEM micrographs of SeNPs. Data on the TEM study of SeSo are not presented, since the “coat” of sorafenib has low contrast.

**Figure 2 cells-12-02723-f002:**
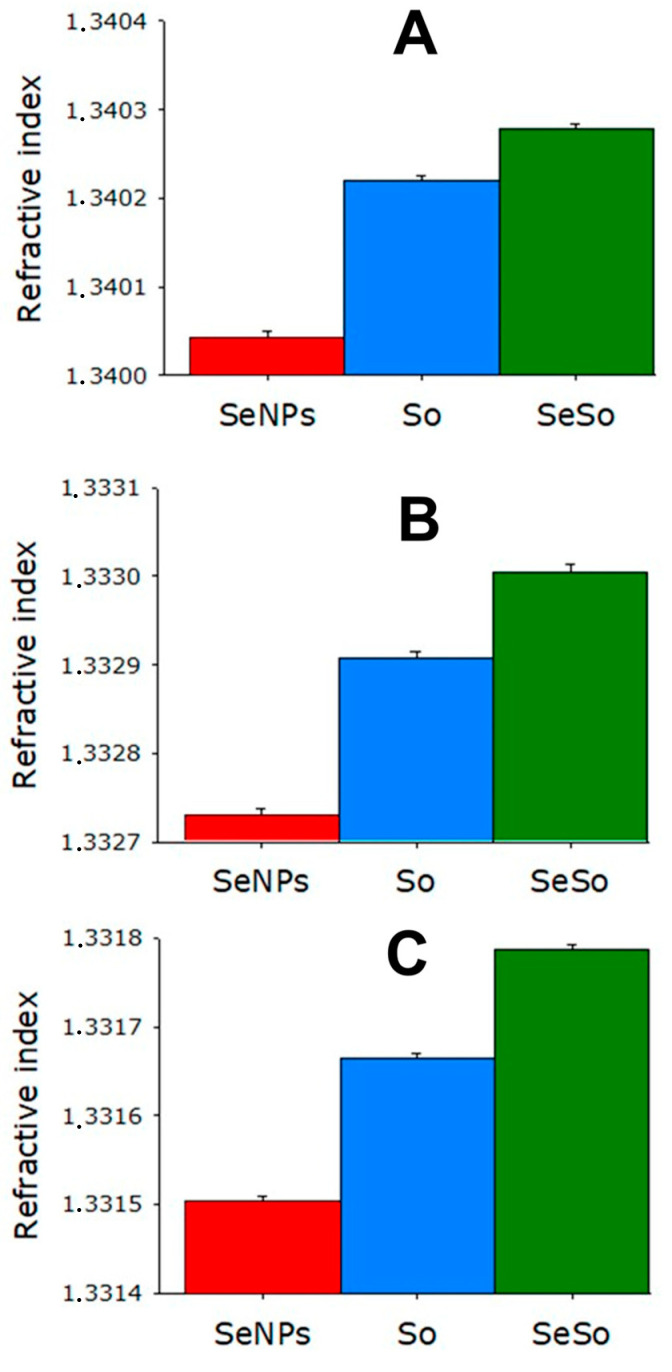
Study of the refractive index of aqueous solutions of SeNPs and SeSo and chemically pure So. The refractive index of the media was calculated using an Abbemat MW (Anton Paar, Austria) precision multi-wavelength digital refractometer Abbemat MW. (**A**) Refractive index measurement at a wavelength of 435.8 nm; (**B**) refractive index measurement at a wavelength of 589.3 nm; (**C**) refractive index measurement at a wavelength of 632.9 nm.

**Figure 3 cells-12-02723-f003:**
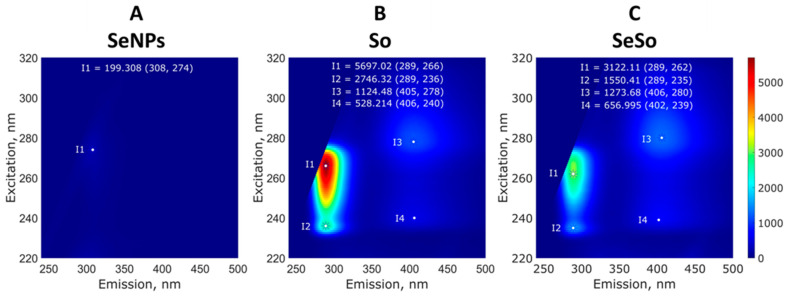
Study of the fluorescence of aqueous solutions of SeNPs (**A**), So (**B**), and SeSo (**C**). Typical 3D spectra are shown; with repeated measurements, the intensity maxima change by no more than a few percentage points. The fluorescence of the samples was studied on a FP-8300 spectrometer (JASCO Applied Sciences, Canada), and measurements were carried out with the shutter turned on in quartz cuvettes with an optical path length of 10 mm at room temperature (~22 °C). Each sample was measured three times.

**Figure 4 cells-12-02723-f004:**
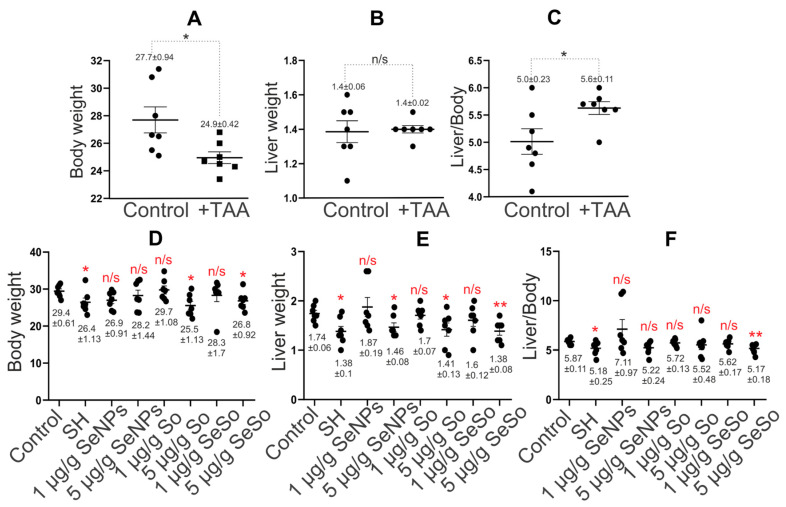
Statistical analysis of liver and animal weight and their ratio. (**A**) Animal weight (g); (**B**) liver weight (g); (**C**) ratio of liver weight to animal weight (%) before and after treatment of animals with TAA (150 µg/g); (**D**) animal weight (g); (**E**) liver weight (g); (**F**) ratio of liver weight to animal weight (%) after intraperitoneal injections of SeNPs, So, and SeSo at concentrations of 1 and 5 µg/g and self-healing animals (SH) in which liver cell regeneration was tested after TAA injections without any therapy. The numbers indicate mean ± SD. Statistical analysis was performed using the unpaired nonparametric *t*-test with the Mann–Whitney test. Ranks were compared. Reliability comparisons were completed relative to the control group. N/s—data not significant (*p* > 0.05), * *p* < 0.05, ** *p* < 0.01. The number of animals in each group was 7.

**Figure 5 cells-12-02723-f005:**
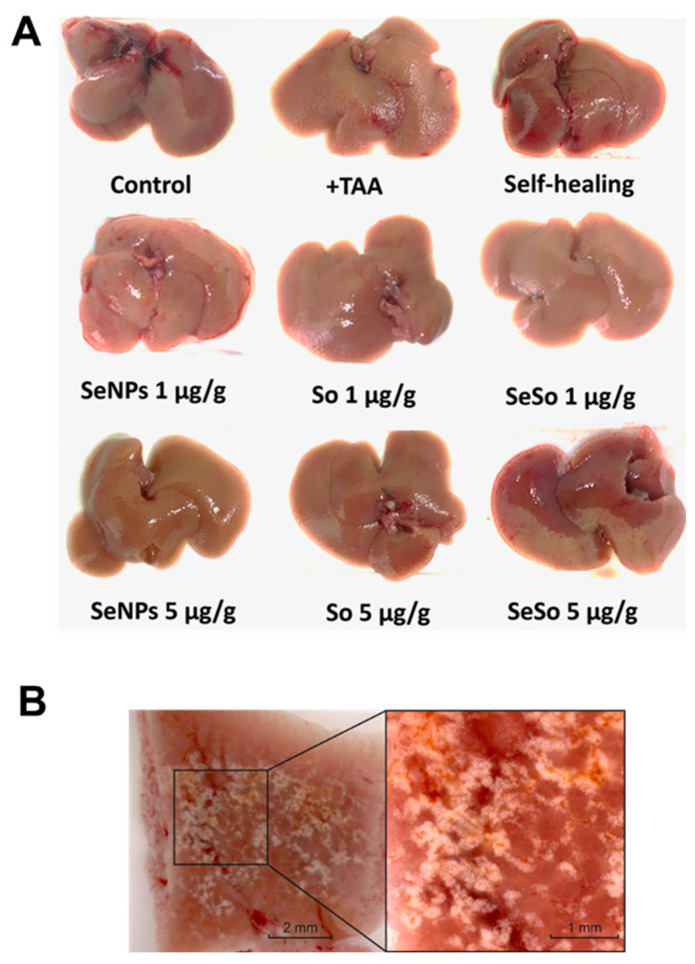
Macroscopic analysis of various liver samples from C57BL/6J mice. (**A**) Photographs of mouse livers before and after injections of TAA, non-particles, and sorafenib at various concentrations, as well as in the self-healing group; (**B**) photographs of a section of mouse liver after TAA injections.

**Figure 6 cells-12-02723-f006:**
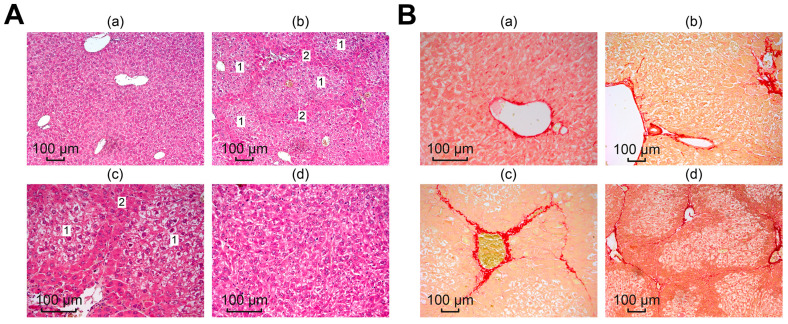
Microscopic analysis of histological liver samples. (**A**) Liver samples stained with hematoxylin and eosin: (**a**) morphology of the liver of the control group; (**b**–**d**) liver morphology after TAA injections, where (1) is false lobules and (2) is fibrous tissue; (**B**) liver samples stained with picro-sirius red: (**a**) morphology of the liver of the control group; (**b**–**d**) liver morphology after TAA injections.

**Figure 7 cells-12-02723-f007:**
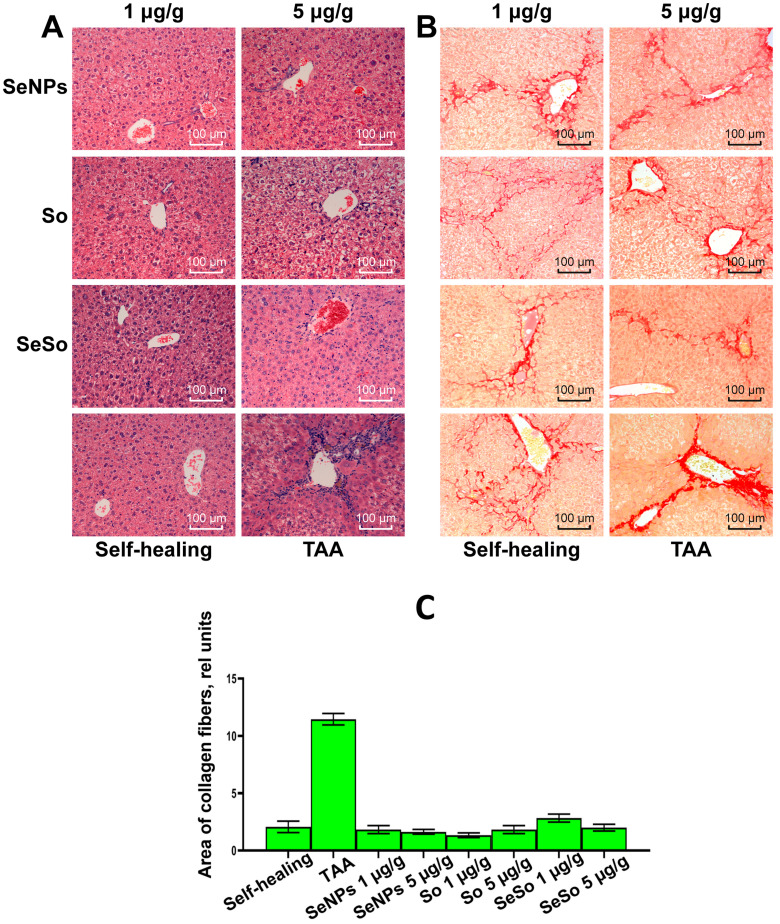
Microscopic analysis of histological samples of mouse liver in different study groups. (**A**) Staining with hematoxylin and eosin; (**B**) picro-sirius red stain; (**C**) the relative area of collagen fibers, calculated using the ImageJ program according to the specified calculation method (https://imagej.nih.gov/ij/docs/examples/stained-sections/index.html, accessed on 15 January 2023).

**Figure 8 cells-12-02723-f008:**
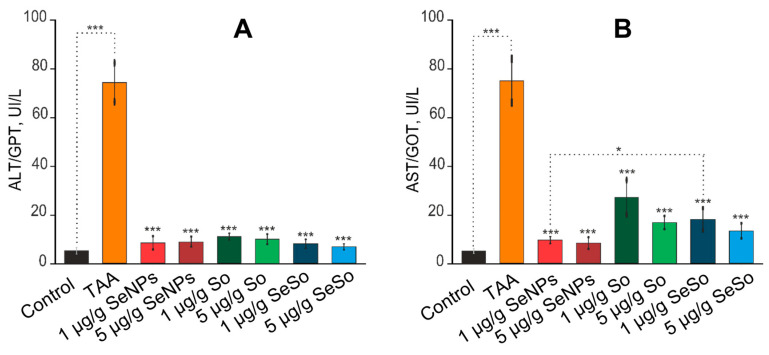
Analysis of the activities of liver enzymes ALT/GPT (**A**) and AST/GOT (**B**) in the serum of the experimental animals. Enzyme activity analysis was performed using the Reitman–Frankel colorimetric method. the standard curve was plotted by using the OD value of the standard and corresponding Carmen units (0, 28, 57, 97, 150, 200 Carmen units) as the x-axis and y-axis, respectively. The standard curve was created with graph software (or EXCEL). The Carmen units of the sample were calculated according to the formula based on the OD value of sample, *** for *p* < 0.001, * for *p* < 0.05

**Figure 9 cells-12-02723-f009:**
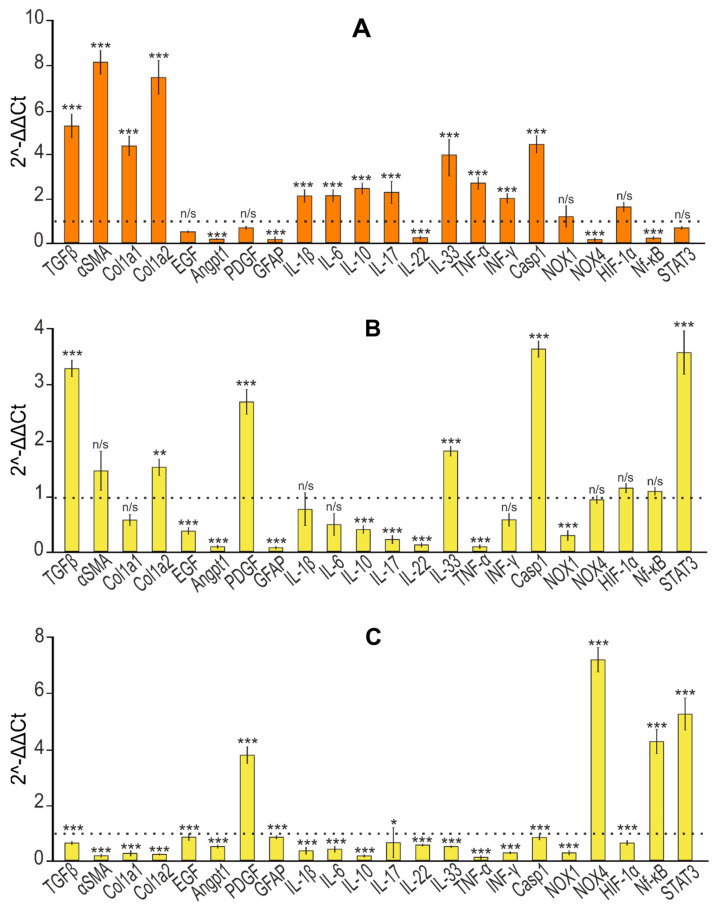
Relative levels of mRNA expression of pro- and anti-inflammatory and pro- and anti-fibrotic genes obtained by real-time PCR. (**A**) After TAA injections in relation to control; (**B**) effect of self-healing in relation to control; (**C**) effect of self-healing in relation to TAA; (**D**) after injections of SeNPs (1 and 5 µg/g) relative to control; (**E**) after injections of SeNPs (1 and 5 µg/g) relative to TAA; (**F**) after injections of So (1 and 5 µg/g) relative to control; (**G**) after injections of So (1 and 5 µg/g) relative to TAA; (**H**) after injections of SeSo (1 and 5 µg/g) relative to control; (**I**) after injections of SeSo (1 and 5 µg/g) relative to TAA. Mean values ± standard errors (SEs) were determined by analyzing data from at least three independent experiments and are indicated by error bars; n/s—data not significant; (*p* > 0.05), * *p* <0.05, ** *p* < 0.01, *** *p* < 0.001.

**Figure 10 cells-12-02723-f010:**
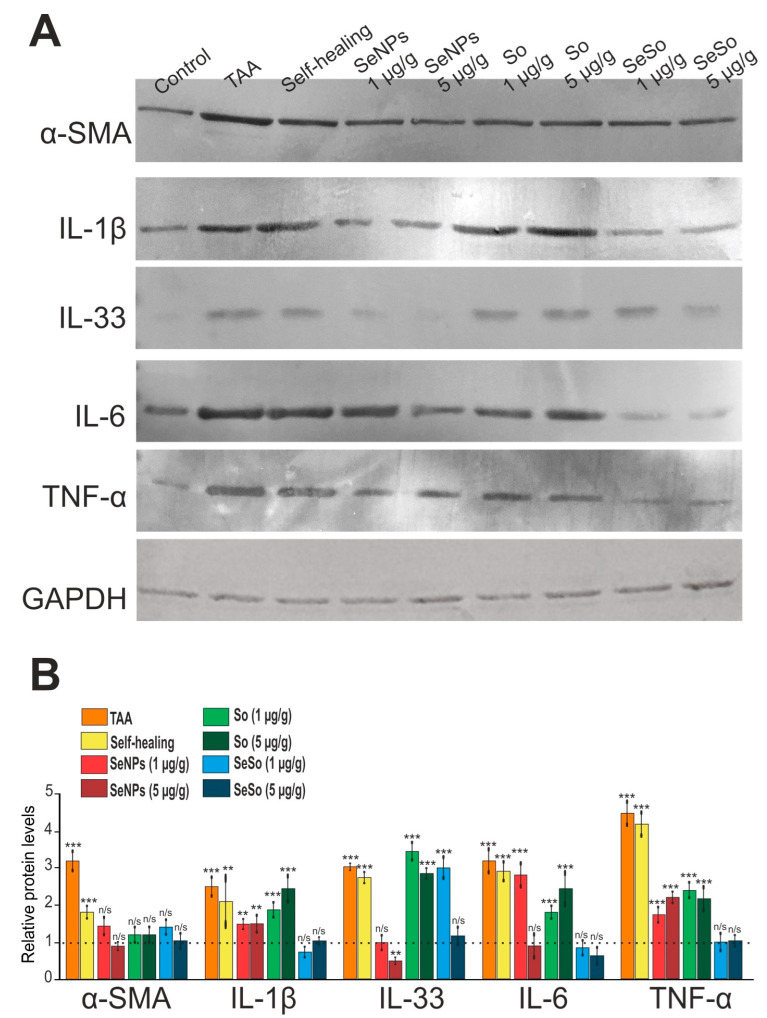
Relative levels of protein quantification in the liver. (**A**) Results of immunoblotting; (**B**) quantification of the studied proteins in the samples obtained using ImageJ software, presented as mean ± standard deviation of three independent experiments. GAPDH was used as a control for normalization; n/s—data not significant; (*p* > 0.05), ** *p* < 0.01, *** *p* < 0.001.

**Figure 11 cells-12-02723-f011:**
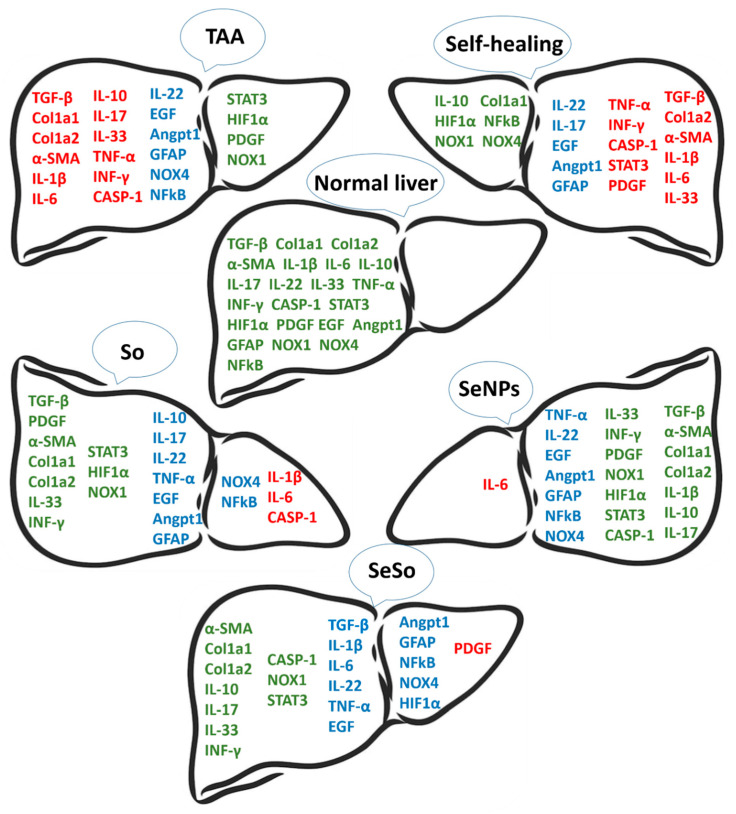
Schematic representation of how the levels of mRNA expression of the profibrotic and proinflammatory genes changed in the livers of all groups of animals relative to healthy animals. Here, green indicates genes whose mRNA expression levels are close to normal levels, red indicates genes whose mRNA expression levels are higher than normal, and blue indicates genes whose mRNA expression levels are lower than normal.

**Figure 12 cells-12-02723-f012:**
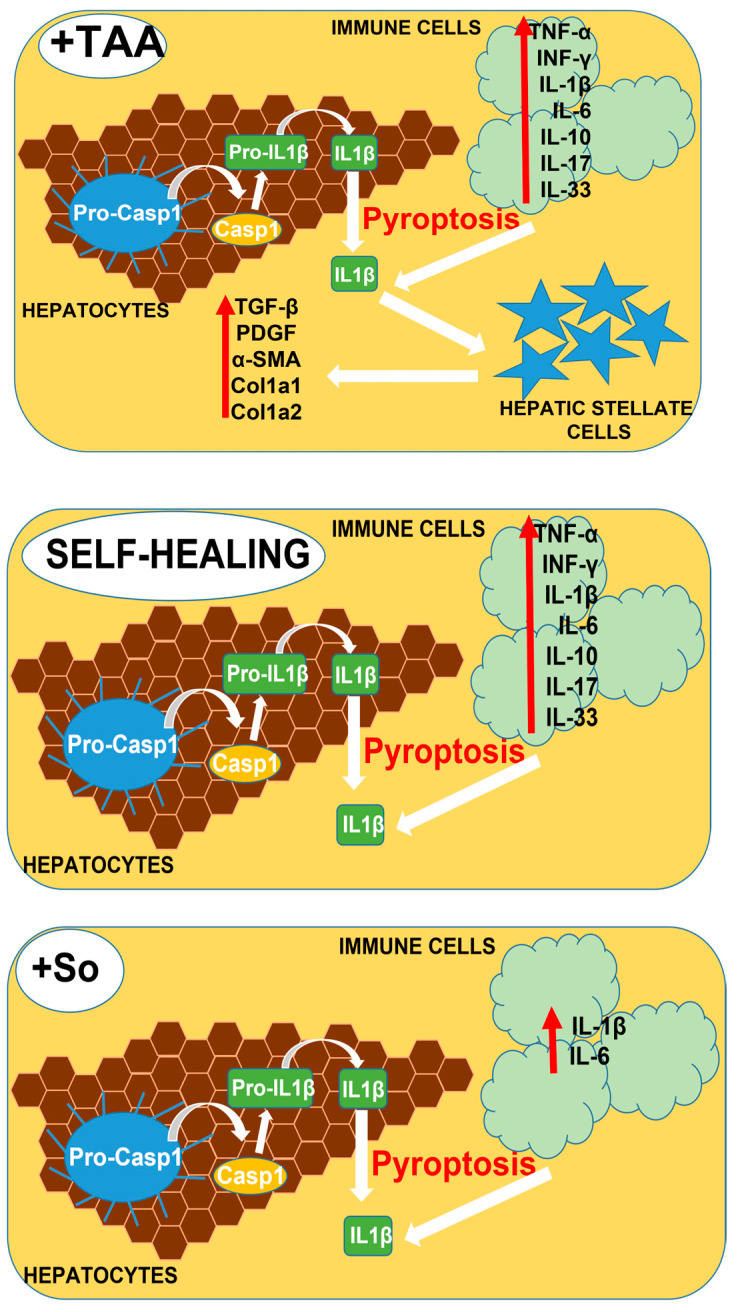
Schematic representation of the processes hypothesized to occur in the liver following TAA, SeNPs, So, and SeSo injections. Exposure of liver cells to TAA leads to an acute inflammatory response accompanied by the formation of the inflammasome, activation of caspase-1, and subsequent activation of IL-1β. These processes occur both in various immune cells and in hepatocytes, which, along with the growth of other pro-inflammatory cytokines, can lead to pyroptosis or cell necrosis. IL-1β, released into the extracellular environment, can contact receptors on the surface of liver stellate cells, activating them. This, in turn, is accompanied by excessive deposition of extracellular matrix and the growth of α-SMA, Col1a1, Col1a2, and EGF, which leads to liver fibrosis. Despite the well-observed trend towards a decrease in pro-fibrotic genes, after injections of So and in the livers of self-healing animals, high levels of expression of CASP-1 and IL-1β still remained, which increases the risks of maintaining cell pyroptosis, which was not observed after similar injections of selenium-containing nanoparticles.

**Figure 13 cells-12-02723-f013:**
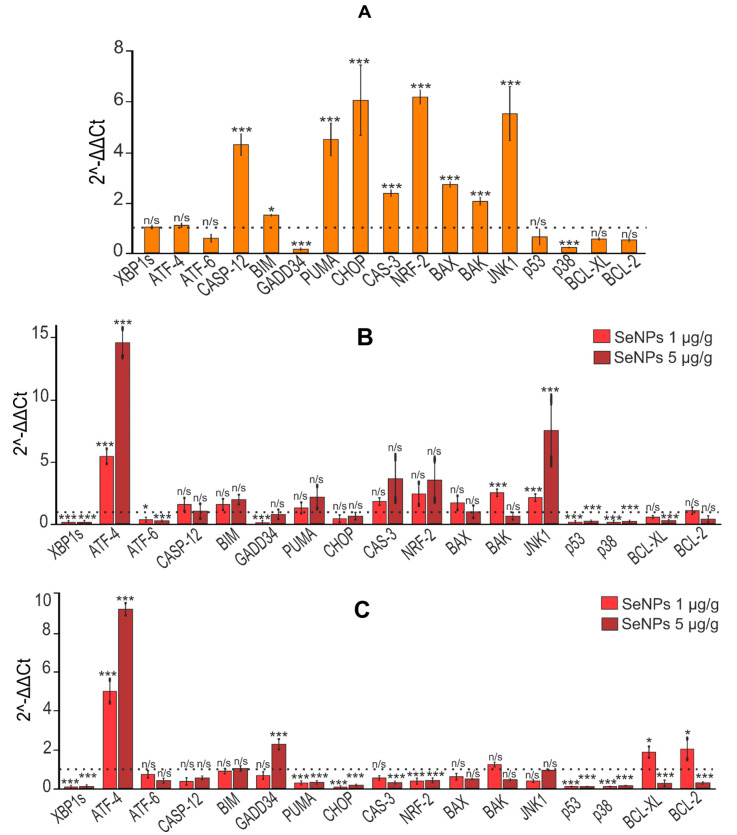
Relative levels of mRNA expression of ER-stress markers obtained by real-time PCR. (**A**) After TAA injections in relation to control; (**B**) after injections of SeNPs (1 and 5 µg/g) relative to control; (**C**) after injections of SeNPs (1 and 5 µg/g) relative to TAA; (**D**) after injections of So (1 and 5 µg/g) relative to control; (**E**) after injections of So (1 and 5 µg/g) relative to TAA; (**F**) after injections of SeSo (1 and 5 µg/g) relative to control; (**G**) after injections of SeSo (1 and 5 µg/g) relative to TAA. Mean values ± standard errors (SEs) were determined by analyzing data from at least three independent experiments and are indicated by error bars; n/s—data not significant; (*p* > 0.05), * *p* < 0.05, ** *p* < 0.01, *** *p* < 0.001.

**Figure 14 cells-12-02723-f014:**
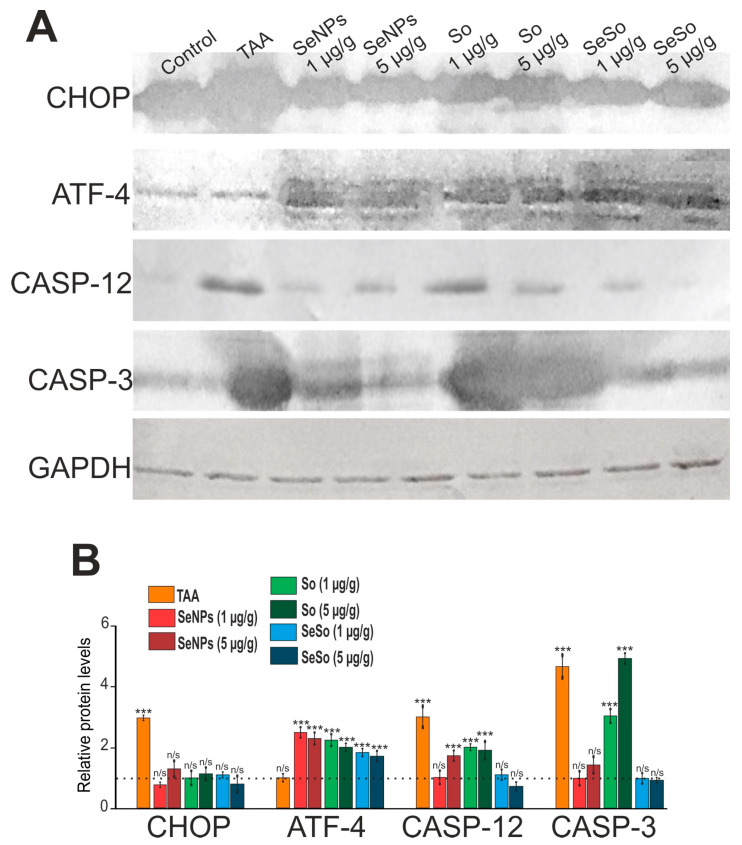
Relative levels of protein quantification in the liver. (**A**) Results of immunoblotting; (**B**) quantification of the studied proteins in the samples obtained using ImageJ software presented as the mean ± standard deviation of three independent experiments. GAPDH was used as a control for normalization; n/s—data not significant; (*p* > 0.05), *** *p* < 0.001.

**Figure 15 cells-12-02723-f015:**
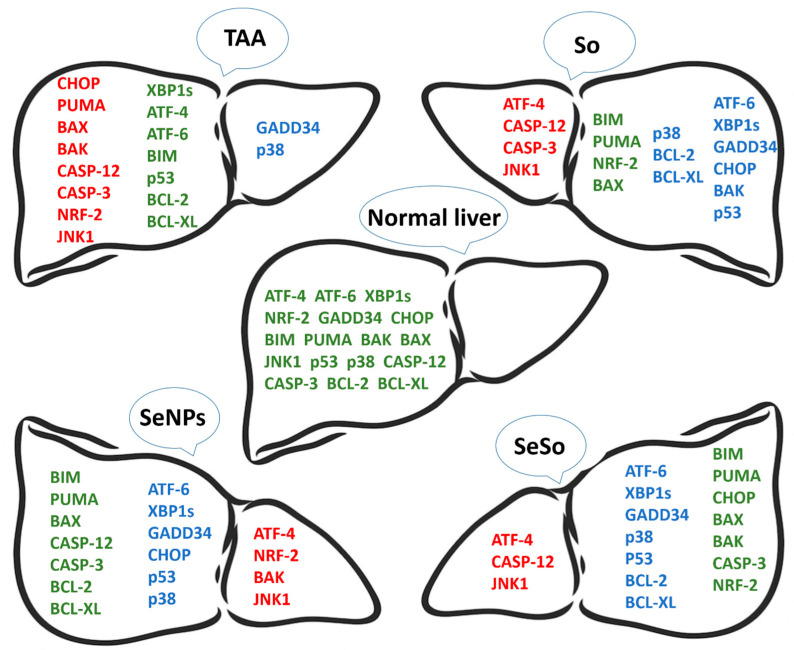
Schematic representation of how the levels of mRNA expression of the studied genes changed in the livers of all groups of animals relative to healthy animals. Here, green indicates genes whose mRNA expression levels are close to normal levels, red indicates genes whose mRNA expression levels are higher than normal, and blue indicates genes whose mRNA expression levels are lower than normal.

**Figure 16 cells-12-02723-f016:**
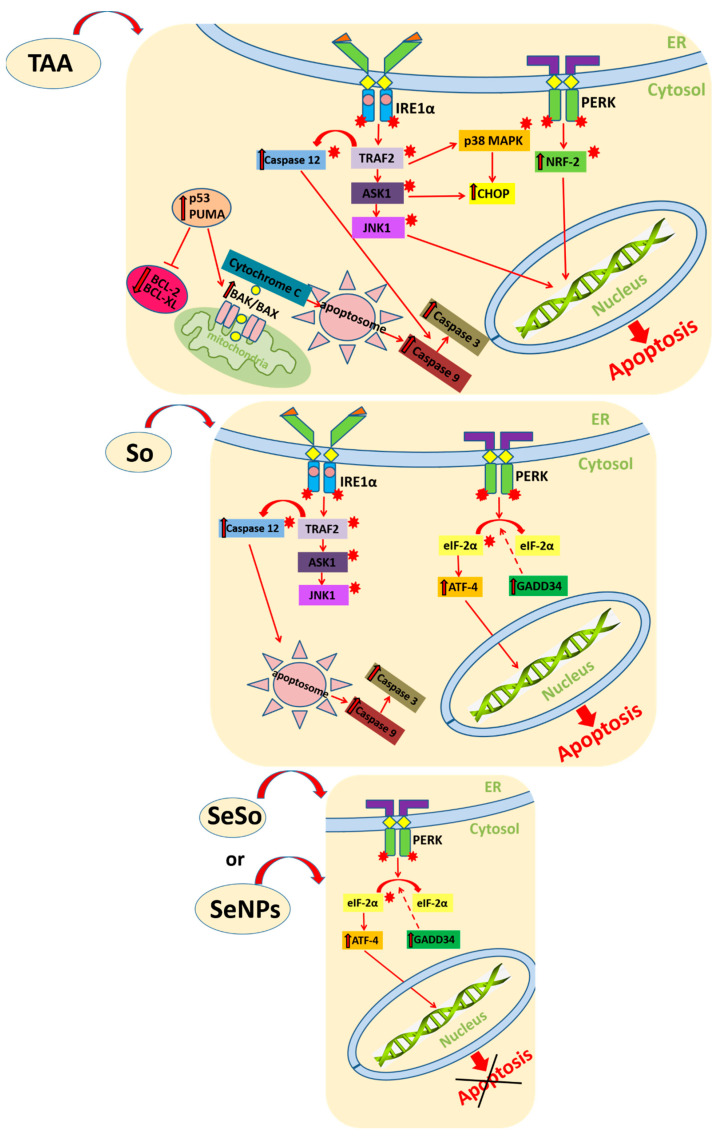
Schematic representation of the activation of UPR signaling pathways during ER-stress, caused by TAA injections, in liver cells. TAA causes activation of IRE1α and PERK UPR pathways through increased expression of JNK1 and NRF-2, which are targets of the IRE1α and PERK kinases. Increased expression of PUMA and p53 leads to increased expression and translocation to mitochondria of the pro-apoptotic proteins BAK and BAX, which ultimately leads to apoptosome formation and activation of caspase-9 and caspase-3. In turn, active caspase-12 also activates effector caspase-3 through activation of caspase-9, which forms the apoptosome. Thus, under conditions of prolonged ER-stress caused by long-term action of TAA, pro-apoptotic signaling pathways are activated. After So injections, liver cells still retain high levels of some pro-apoptotic genes: CASP-12, CASP-3, and JNK1, which may indicate a high risk of apoptosis in liver cells. After injections with SeNPs or SeSo nanoparticles, the adaptive UPR pathway is activated through the PERK signaling pathway. Thus, SeNPs and SeSo are able to neutralize the effects of ER-stress and lead to a high increase in the expression of mRNA of a number of pro-apoptotic genes caused by TAA injections.

**Table 1 cells-12-02723-t001:** Sequences of oligonucleotides used in the real-time PCR reaction.

Gene Names	Forward Primer 5′- > 3′ Reverse Primer 5′- > 3′	Gene Names	Forward Primer 5′- > 3′ Reverse Primer 5′- > 3′
*TGF-β*	ATGCAATGGGCTTAGTGTTCTG TCCTGTTGGCTGAGTTGTGAC	*HIF-1α*	GGCGACTGTGCACCTACTATG TGATCCAAAGCTCTGAGTAATTC
*EGF*	CCTTGGTTTGTGGTCCTAGAG CTGGGGTCCTCTGTCACTTG	*XBP1s*	AGTCCGCAGCACAGCAGGT AGAGAAAGGGAGGCTGGTAAG
*Col1a1*	CATCACCTATCACTGCAAGAACAGGTCTTGGTGGTTTTGTTATTC	*ATF-4*	TCGGGTTTGGGGGCTGAAG AAACAGAGCATCGAAGTCAAAC
*Col1a2*	TCTCAGAACATCACCTACCACCACGGAATTCTTGGTCAGCAC	*ATF-6*	AGGAGGGGAGATACGTTTTAC CGAGGAGCTTTTGATGTGGAG
*Angpt1*	GGGACAGCAGGCAAACAGAG CTGGGCCCTTTGAAGTAGTG	*CASP-1*	AGAGAAATGAAGTTGCTGCTGG ATCACCTTGGGCTTGTCTTTC
*PDGF*	ATTGAGATTGTGCGAAAGAAGC GGGGGCAATACAGCAAATACC	*CASP-3*	CTCTTCATCATTCAGGCCTGC GACCCGTCCTTTGAATTTCTC
*α-SMA*	AGGAAGGATCTCTATGCTAACAAC ACTTAGAAGCATTTGCGGTGG	*CASP-12*	TGTTGGTGTTATCATTTGGAGG TTTTCTTTTCTTCTCAGCTACAG
*GFAP*	CCGCCACCTGCAGGAGTAC TGTATTGTGAGCCTTTTGAGAG	*BIM*	AATGGCCGGCTATGGATGATG GCCAATTGGGTTCACTGTCTG
*IL-17*	CCCTCAAAGCTCAGCGTGTC CCAGCTTTCCCTCCGCATTG	*GADD34*	GAGTCCCATGAAGAGATTGTAC ACCAGCCCAGCAGCCACTTAG
*IL-1β*	CGTGCTGTCGGACCCATATG GCTCTTGACTTCTATCTTGTTG	*PUMA*	TGAAGATCTGCGCCGGGAG GAGAGGGACATGACGCGTG
*IL-33*	TTTTGGAGAATGGATGTTATGTG TTTGTGAAGGACGAAGAAGGC	*CHOP*	CAGCTGGGAGCTGGAAGCCTG GACCACTCTGTTTCCGTTTCC
*IL-22*	GCTCCCCCAGTCAGACAGG TAGAAGGCAGGAAGGAGCAG	*BAX*	TAAAGTGCCCGAGCTGATCAGAAC CTTCCCAGCCACCCTGGTCTT
*IL-10*	AGCATGGCCCAGAAATCAAGG AGACTCAATACACACTGCAGG	*BAK*	CAGATGGATCGCACAGAGAG GCGTCTTTGCCCTGGGGAG
*IL-6*	TCCAGAGATACAAAGAAATGATG TTGGAAATTGGGGTAGGAAGG	*BCL-2*	AAGTCAACACAAACCCCAAGTCCTC GCAGATCTTCAGGTTCCTCCTGAGA
*NFkB*	TTAAAGAAACACTCAACAGCCAG TTCAGCACTCGCACGGACAC	*BCL-XL*	AGAGTGAGCCCAGCAGAACC GCAAGTTGGATGGCCACCTATC
*STAT3*	CCCCGTACCTGAAGACCAAG ATGGGGTTCGGCTGCTTAGG	*NRF-2*	CACATTGGGATTCACGCATAGGAGCACT TCCTGGACGGGACTATTGAAGGCTG
*TNF-α*	TGGAAAGACAGAGGGTGCAG TTGTCCCTTGAAGAGAACCTG	*JNK1*	AGAAGCAGAAGCCCCACCAC ACTGCTGTCTGTATCCGAGG
*INF-γ*	GTGACATGAAAATCCTGCAGAG TGAGGCTGGATTCCGGCAAC	*P53*	TGTTTAGGTCAAGGTGTCTCC GAACACAGCCCCTAACACAG
*NOX1*	ACAAGAGATGGAGGAATTAGG TTCCTAGGATCCAGACTCGAG	*P38*	GTCGACCTACTGGAGAAGATC AGTGAGATAGACAGAACAGAAAC
*NOX4*	TACCTCAGTCAAACAGATGGG TGTCCCATATGAGTTGTTCCG	*GAPDH*	GTAAAGACCTCTATGCCAACAC GGTGCACGATGGAGGGGC

## Data Availability

The data presented in this study are available on request from the corresponding authors.

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
