# Peer review of "Antifibrotic Effect of Selenium-Containing Nanoparticles on a Model of TAA-Induced Liver Fibrosis"

_cells, 2023, doi:10.3390/cells12232723_

Round 1

Reviewer 1 Report

Comments and Suggestions for Authors

This paper reported selenium nanoparticles in their anti-fibrotic, anti-inflammatory, and anti-apoptotic effects. Experiments were well-designed to compare gene/protein expression, and also in mouse fibrosis TAA model. Results showed promising treatment effects alone or in the form of selenium and sorafenib nanocomplex.

Overall it provides a lot of information on this material in the treatment of popular diseases.  I have no major concerns. The paper could be improved in most of the figures. Here are my comments.

1.       When TAA is first mentioned (in the abstract), the full name thioacetamide should be given.

2.       In the figure legend, there lack of full description, for example, in Fig. 1, no explanations of what are A, B, C, and D.

3.       In Figure 2 and afterward, the arrangement may appear better if in the order of “SeNP, So, SeSo” as SeSo are combined drug and nanoparticle. Again, A, B, and C should be explained.

4.       Fig. 4 needs labeling of A, B, C, D

5.       Fig. 8 needs to be reformatted- font lost format.

6.       In Fig. 11,  I suggest using arrows to indicate the up and down expression instead of different colors in order to make it easier to view.

7.       Fig 12, a Schematic representation of the processes hypothesized to occur in the liver following TAA SeNPs, So, and SeSo injection, is really difficult to comprehend. Such a scheme is supposed to make readers understand easier instead of making it harder. I suggest redesigning it to make it simpler.

8.       Minor grammar mistakes, e. g “Results and its discussion” should be “results and discussions”.

Comments on the Quality of English Language

Grammar and word choices can be improved.

Author Response

Dear Reviewer!

The authors thank you for your interest in our work and valuable comments and suggestions. We have tried to correct these comments.

  1. In the “Abstract” section we have included the full name of thioacetamide (TAA).
  2. In the captions to the pictures they wrote in more detail what the pictures mean. under the letters A, B, C..
  3. In Figure 2, the columns have been swapped
  4. In Fig. 4 we added a letter designation to each figure and added a description to the legend
  5. We reformatted Fig. 8
  6. If you don’t mind, we would leave Fig. 11 in the form it is now, since earlier, when writing the manuscript, we made a version with arrows instead of different colors, but visually this scheme is less perceived. Moreover, there are many genes in the scheme, the expression of which did not change significantly.
  7. We have simplified as much as possible, Fig. 12
  8. We checked the text for typos and errors, in particular, we made a correction in the title “results and discussions”

Reviewer 2 Report

Comments and Suggestions for Authors The manuscript "Antifibrotic effect of selenium-containing nanoparticles on a model of TAA-induced liver fibrosis" presents a wide range of results related to liver fibrosis, but also the effects of substances that trigger this process in the opposite direction. More specifically, the effects of selenium-containing nanoparticles (SeNPs), which are widely used in medicine from diagnosis to therapy, SeNPs in combination with sorafenib, and sorafenib itself, which is already used in therapy, were examined. SeNPs and sorafenib-doped nanoparticles are obtained by laser ablation, a rapid method that provides the final product in high yield and is referred to as a "green" method because no toxic chemical precursors are used to synthesize nanoparticles. In the Results and their discussion section, the results of TAA-induced liver fibrosis and the results after experimental treatments are presented extremely clearly and concisely. The basic physiological parameters, the detailed qualitative and quantitative histological analyses, the gene expression modes of 40 genes (including genes that have a fibrotic effect, genes that affect stellate cell activation, genes that control the expression of numerous cytokines and transcription factors...), and the expression of the corresponding proteins are presented. Despite the huge amount of data, the text is clear and unambiguous, the accompanying diagrams are appropriate, and the schematics presented summarize the information in a very clear and informative manner. From a medical point of view, the results of the self-healing group are also very interesting, showing a pro-inflammatory state that persists after TAA removal, which explains why fibrosis often progresses to cirrhosis in untreated cases. The presumed mechanism of action of TAA involves activation of caspase-1 and the inflammatory response of the liver, which may lead to pyroptosis of hepatocytes, while SeNP and the combination of SeNP and sorafenib induce changes in the expression pattern of numerous genes that have an anti-inflammatory effect, thereby significantly reducing the adverse effects induced by the use of TAA. Finally, the authors summarize the events at the cellular level and conclude, based on the expression of numerous markers, that TAA triggers ER stress activating pro-apoptotic signaling cascade to the cytosol, nucleus and mitochondria, leading to apoptotic death of hepatocytes. More specifically, the IRE1alpha and PERK signaling pathways of the UPR are determined to be activated by TAA. Both SeNP and SeNPs in combination with sorafenib are able to neutralize the negative changes induced by TAA affecting proapoptotic signals in the opposite direction. The data obtained in this study are relevant and will certainly contribute to the future use of SeNP (alone or as a drug carrier) in clinical practice.  

Based on the above, the opinion of this Reviewer that the manuscript should be accepted after only a few minor requests:

Abbreviations should be introduced and the rest of the text (ER-stress, TAA ...).

INTRODUCTION

Pg. 2, line 2 correct sorafnib to sorafenib

MATEIAL AND METHODS

2.3. Injection protocol: The presence of an appropriate control group should also be indicated (one sentence). Pg. 4 Is it possible to express the concentration of nanoparticles also as mass per volume?

Mandatory to add section Statistical analysis in Material and methods.

RESULTS AND THEIR DISCUSSION

Insert bar in figure 1D.

In 3.2. section The weight of animals... second paragraph, according to Figure 4B (body weight) So at concentration of 5mg decreased average weight by 4g (not 1mg).

In the third paragraph: there is no unit of measurement for the ratio of liver weight to animal weight, and corrections should be made in the text accordingly.

Next paragraph: As can be seen from the graph, the body weight did not reach the control value at the So concentration of 5mg/g, it is still significantly reduced, but at the concentration of 1mg/g there is no significant differences compared to control, should be corrected.

What is the concentration of the nanoparticles used in Figure 6A(d)?

It would be good to add the self-healing group in Figure 8, simply because this group also appears in the other results.

Pg. 15 last paragraph: I think it would be good to underline the results presented above by stating - strongly contribute to...or trigger the development of liver fibrosis. Pg. 17, fourth paragraph: since all treatments significantly affect STAT3 expression, I would delete part of the sentence this indicator was close to the control value. In the section 3.6. Since this is the Results and discussion section, there should be a brief introduction (few sentences) that refers to ER stress and activation of an unfolded protein response (UPR) signaling network in the context of the research. (Introduce abbreviations in this part as well).

Author Response

Dear Reviewer!

The authors thank you for your interest in our work and valuable comments and suggestions. We have tried to correct these comments.

  1. We have included a decoding of the abbreviations for TAA and ER-stress in the “Abstract” section
  2. In the Introduction section, in paragraph 2, line 2, a correction was made sorafnib to sorafenib
  3. In the section Materials and methods, paragraph 2.3. inserted information about the control group.
  4. Inserted statistical processing of results into the Materials and Methods section
  5. We inserted a bar into Fig. 1D
  6. A correction was made to section 3.2.: 1 µg/g was corrected to 5µg/g for sorafenib.
  7. We have corrected all the errors you indicated in section 3.2.
  8. In Figure 6 shows histological sections of the liver only after TAA injections; we have made appropriate corrections to the legend to the figure.
  9. Unfortunately, we do not have data on the self-healing group in experiments measuring ALT/AST levels in the blood. If you don't mind, we would leave these data unchanged.
  10. Corrections have been made in the last paragraph on page 15.
  11. Corrections have been made in the text regarding the expression of STAT3 gene mRNA (p. 18, paragraph 3).
  12. At the beginning of the Results and Discussion section, we inserted additional information as you suggested.

Round 2

Reviewer 1 Report

Comments and Suggestions for Authors

This manuscript has been significantly improved with accepted publishing quality.

Author Response

Thanks for reviewing our article

Reviewer 2 Report

Comments and Suggestions for Authors

1) The  authors must make a corrections in the text related to the ratio of liver weight to total weight of animal (Pg. 9 and 10): namely, this parameter does not have a unit and the expression in grams is not appropriate. Also, differences between groups regarding this parameter cannot be expressed in grams. They can be expressed as percentages or how many times parameters differ between groups.

2) For Figure 6B add: (b)-(d) liver morphology in the TAA group.

Author Response

Dear Reviewer,
The authors of the article are grateful to you for your comments; we have corrected them in the text and marked them with a green marker.